# Safety, Feasibility, and Effects of Short-Term Calorie Reduction during Induction Chemotherapy in Patients with Diffuse Large B-Cell Lymphoma: A Pilot Study

**DOI:** 10.3390/nu13093268

**Published:** 2021-09-19

**Authors:** Chia-Chun Tang, Tai-Chung Huang, Feng-Ming Tien, Jing-Meei Lin, Yi-Chen Yeh, Ching-Yi Lee

**Affiliations:** 1School of Nursing, College of Medicine, National Taiwan University, Taipei 100025, Taiwan; chiatang@ntu.edu.tw (C.-C.T.); alchemist19940520@gmail.com (Y.-C.Y.); 2Division of Hematology, Department of Internal Medicine, National Taiwan University Hospital, Taipei 100225, Taiwan; tch01@ntu.edu.tw (T.-C.H.); b92401007@ntu.edu.tw (F.-M.T.); 3Department of Dietetics, National Taiwan University Hospital, Taipei 100225, Taiwan; kyomilin@ntuh.gov.tw; 4Department of Nursing, National Taiwan University Hospital, Taipei 100225, Taiwan

**Keywords:** short-term calorie reduction, fasting, cancer, lymphoma, chemotherapy, calorie restriction

## Abstract

Short-term calorie reduction (SCR) requires individuals to reduce their calorie intake to less than 50% of normal requirements and has shown good tolerance and potential benefits in prior studies addressing gynecological cancer patients. More studies are needed to further confirm its safety, feasibility, and effects in patients with different cancers, including hematological malignancies. This pilot cohort study with a matched-pair comparison group was registered at ClinicalTrails.gov [201810112RIND]. Adult patients diagnosed with advanced-stage diffuse large-B cell lymphoma were recruited (SCR group) and matched with one comparison patient (comparison group), each in a manner blinded to their outcomes. The SCR group undertook at least two cycles of 48 h water fast along with their chemotherapy R-CHOP. Descriptive analysis and generalized estimating equations were used to analyze the data. Six participants completed multiple cycles of SCR and were compared to their six counterparts in the comparison group. The results showed that SCR is safe and feasible in terms of a high compliance rate and stable nutritional status. The SCR was associated with benefits in post-chemotherapy hematological parameters (i.e., erythrocyte [*p* < 0.001] and lymphocyte counts [*p* < 0.001]). More randomized controlled trials are needed to validate the effects of SCR on different types of cancer populations.

## 1. Introduction

In the last 10 years, calorie reduction has been suggested as a promising nutritional intervention in adjuvant therapy for standard cancer treatment [1,2]. Based on the proposed mechanisms of calorie reduction, such as enhancing tumor cell sensitivity to chemotherapy, decreasing inflammation, increasing circulation of T cells, inhibiting neovascularization, and promoting tissue regeneration, calorie reduction may not only promote therapeutic effects but also mitigate the side effects of chemotherapy for cancer patients [2,3,4,5,6,7]. However, performing long-term calorie reduction and detecting its effects is time consuming, limiting its clinical feasibility. Additionally, long-term calorie reduction is likely to result in weight loss, compromising patients’ adherence [5] and causing worries of malnutrition in patients with cancer. Accordingly, short-term calorie reduction (SCR) with a higher calorie reduction but shorter implementation duration has emerged as a novel strategy to enhance the effect of chemotherapy and reduce toxicities [3,8,9].

In general, SCR reduces one’s normal calorie intake to below 50% for no more than one week and is arranged alongside chemotherapy. In contrast to long-term calorie reduction, SCR shows comparable safety but better tolerance and more rapid effects enhancing the care and treatment of cancer patients [10]. Specifically, SCR has been demonstrated to have good early adherence [11,12,13,14,15,16], with self-limiting side effects [11,17,18,19], minor body weight loss [11,17,18,19,20], no adverse effects on nutrition and metabolism status [12,17], and preserved lean body mass [17]. Biologically, SCR is believed to reduce side effects [11,12,17,18,19,21] but inversely enhances the therapeutic effects of chemotherapy [17,18,19]. The SCR is proposed to have protective or regenerative effects on normal cells in patients who received chemotherapy [9], which helps stabilize serum glucose [22], preserve hematological function [12,17,18,19], and alleviate DNA damage [12,17,21]. With respect to enhancing treatment response, SCR may inhibit tumor growth, cellular proliferation, and metabolism by sensitizing tumor cells to chemotherapy [3,8,9]. It also helps ameliorate the complications of cancer [23]. From a psychological perspective, SCR improves subjective wellbeing [11,18,20,23], mitigates illness perception [24], increases adherence to chemotherapy [25], and enhances self-control by reducing uncertainty [25].

Although current evidence suggests that SCR is a promising adjuvant nutritional strategy, inconsistent and inadequate evidence prevents SCR from being endorsed in treatment guidelines as an official recommendation [13,15,26,27]. First, most studies that address the safety and effectiveness of SCR in cancer patients have used heterogeneous cancer patients, which hinders the examination of specific effects from SCR [17,18,28]. Studies that focused on homogeneous cancer types were conducted in female patients with breast and ovarian cancers [11,12,19,20,21,23,24,25], with a couple being carried out in glioma patients [29,30]. Nevertheless, both sexes may have a differential preference for behavioral modification, such as men’s preference for exercise and water intake and women’s preference for diet control and reduction of sugar-sweetened beverages in health promotion programs [31,32]. Additionally, the chemotherapy regimens and symptomatology of various cancers may also confound the evaluation of safety and feasibility of SCR. Lastly, food is beyond just a source of nutrition, and diet plays different roles in various cultures [33,34]. Social and cultural dietary factors also influence the implementation of SCR, such as attitudes toward calorie control and food preparation during illness and food sharing in the context of social norms [25]. The present quantitative studies on SCR have been conducted in Western society [11,12,17,18,19,21,25], raising concerns of external validity across ethnicities. Taken together, expanding study participants into different sexes, diagnoses, and ethnicities [35,36] before implementing SCR in overall or specific cancer populations is called for.

Diffuse large B-cell lymphoma (DLBCL), a subtype of non-Hodgkin lymphoma, is an optimal research target. First, while DLBCL is a common type of lymphoma, it is more prevalent in men, which deserves sex-specific nutrition intervention. Second, the induction chemotherapy for DLBCL (i.e., rituximab, vincristine, doxorubicin, cyclophosphamide, and prednisolone; R-CHOP) has been standardized. Short-term hospitalization (2 days) or outpatient chemotherapy visits for R-CHOP is the standard procedure in Taiwan. These consistent procedures and chemotherapy drugs may facilitate adherence to the study protocol and the implementation of SCR. Third, R-CHOP includes regular administration of steroids, which also raises nutritional concerns, such as hyperglycemia [37]. Patients receiving R-CHOP may benefit from SCR in terms of glycemic control. Thus, investigating SCR’s safety and feasibility in Chinese DLBCL patients not only provides solutions to the limitations of previous studies in terms of sex, diagnoses, and ethnicity, but also reduces confounding effects from disease per se and the treatment course. Finally, although about half of the patients with DLBCL can be cured by R-CHOP treatment, the other half do not respond well to this standard treatment [38]. Several factors predict poor treatment outcomes in DLBCL, including advanced stage (Ann Arbor stage III or IV) [39]. Identifying an adjuvant therapy that can enhance the treatment outcome is imperative.

This study aimed to evaluate the safety, feasibility, and effects of repeated application of the SCR protocol in patients with DLBCL undergoing R-CHOP. The research hypotheses were as follows: (1) The SCR is safe and feasible, as determined by the retention rate, compliance rate, patient-reported fasting-related symptoms, and nutritional status. (2) The SCR is better than regular care in terms of protecting normal cells from drug toxicity as measured by hematological indicators before and after SCR.

## 2. Materials and Methods

This was a pilot cohort study with matched-pair analysis. Between October 2020 and July 2021, potential participants were identified and recruited at a hematological medical center in Taipei, Taiwan through physician referrals. About 50 patients with advanced DLBCL were seen by hematologists each year at the participating medical center. The inclusion criteria were patients who were (1) diagnosed with advanced DLBCL (stage III or IV), (2) planned to receive R-CHOP regimen, and (3) 20 years old or older. The reason for focusing on patients with advanced disease was that the effects of SCR may be more obvious in these patients than in patients in early stages of the disease, as they have a poor response to chemotherapy. Patients who (1) had a body mass index (BMI) less than or equal to 18.5, (2) had albumin levels lower than 3.4 g/L, (3) had a history of eating disorders, (4) had difficulties in following the instructions of calorie modifications due to a physiological or psychological condition; (5) had been diagnosed with diabetes mellitus or had physiological or psychological conditions due to which calorie modification may have had a negative effect on their physical or psychological status, (6) had special dietary restrictions, or (7) had only one unfinished chemotherapy cycle upon recruitment were excluded. This study was approved by the ethics committee of the National Taiwan University Hospital. Written consent was obtained from all participants. The matched comparison group was identified through chart review and is described in detail in the following paragraph.

### 2.1. The Short-Term Calorie Reduction (SCR) Intervention

The SCR intervention was established based on a published literature review conducted by the authors [10]. Aligned with the participants’ R-CHOP regimen, they performed water fast for 24 h before day one and 24 h on day one of the R-CHOP cycle. During fasting, participants were not allowed to eat, drink, or use any intravenous fluid containing calories but were encouraged to drink water. Once the participants completed a total of 48 h of fasting, they resumed normal calorie intake until the day before the next cycle of chemotherapy. While each cycle of R-CHOP took 21 days, all participants received a total of six cycles over four to six months. Upon agreeing to participate in this study, participants were required to perform SCR repeatedly and continuously with each remaining chemotherapy cycle. Thus, the participants performed SCR 2–5 times based on the timing of their participation in this study. If needed, participants could take a researcher-provided food package containing food with a maximum calorie count of 250 kcal every 24 h. As part of the calorie restriction, the nutrient contents of each food package were designed according to acceptable macronutrient distribution ranges for carbohydrate (45–65% of calorie intake), protein (10–35% of calorie intake), and fat (20–35% of calorie intake).

### 2.2. Data Collection and Outcome Measurements

Feasibility and safety were evaluated by subject retention rate, compliance rate, and nutritional status (i.e., prealbumin, cell phase angle, and BMI). Specifically, cell phase angle was measured by InbodyS10, which utilizes the techniques of bioelectrical impedance analysis. The effect of SCR was evaluated based on hematologic parameters (i.e., erythrocyte and leukocyte counts). In addition to the baseline, hematological data were collected approximately 21 days after SCR, and nutritional data were collected shortly after the injection of the chemotherapy drug (during SCR). Demographic information, such as disease stage, age, and sex, were obtained upon recruitment. After each SCR, subjects were required to record a food diary for one day to ensure that they resume normal-calorie consumption.

### 2.3. Matched Pair-Analysis

Each participant who received SCR was matched with one comparison patient in a manner blinded to their outcomes. The criteria for matching the pairs were as follows: (1) the same diagnosis (based on ICD-10 and ICD-O-3), (2) the same disease status (primary tumor vs. recurrence), (3) similar tumor stage (advanced stages as determined by stage III and IV in Lugano (modified Ann-Arbor) status), (4) the same sex, (5) similar age (±5 years), and (6) received the same chemotherapy regimen (R-CHOP) at the same hospital during the last five years.

### 2.4. Statistical Analysis

Descriptive statistics were used to describe the data characteristics. To compare nominal and ordinal matching variables between the two groups, including sex and stage, Student’s t-test was used to compare continuous variables between groups, and the Mann–Whitney U test was used to examine differences in ordinal variables between groups. As outcomes were measured repeatedly, a generalized estimation equation (GEE) was used to determine the effects of SCR on hematological indicators while considering within-subject factor correlation and confounding variables. The IBM SPSS 21 (IBM Corp, Armonk, NY, USA) and Microsoft Excel (2016, Microsoft Corporation, Redmond, WA, USA) were used to organize the quantitative data and facilitate the data analysis. The Research Electronic Data Capture (REDCap, Vanderbilt University, Nashville, TN) system was used for data management. All tests were two-tailed, and *p* < 0.05 was set as the cutoff for statistical significance.

## 3. Results

Data are presented as mean ± standard deviation, unless otherwise stated. Nine patients were referred by hematologists and approached; six were included in this study and matched to their six counterparts. Three patients were excluded because of BMI <18.5 [*n* = 2] and deteriorated physical status [*n* = 1]. All 12 patients were male, Chinese, and successfully completed six cycles of R-CHOP consisting of rituximab (375 mg/m^2^ intravenously (i.v.)), cyclophosphamide (750 mg/m^2^ i.v.), doxorubicin (50 mg/m^2^ i.v.), vincristine (1.4 mg/m^2^ i.v.), and prednisolone (60 mg/m^2^/day by mouth for 5 days). Detailed participant characteristics for both groups are displayed in Table 1. No statistically significant differences in the characteristics were observed between the groups. All participants in the SCR group successfully completed at least two cycles of SCR, ranging from two to five cycles. The variations in completed cycles were only affected by the time they agreed to participate in the study. For example, if a subject agreed to participate in the study in his fourth cycle of R-CHOP out of six, then the maximum number of SCR cycles in which he could participate was two. Participants started SCR when they received their second (*n* = 3), third (*n* = 1), fourth (*n* = 1), and fifth (*n* = 1) cycle of chemotherapy. Once the subjects started the intervention, they were able to carry out SCR repeatedly along with all remaining chemotherapy cycles (*n* = 6, 100%). The recruitment and intervention processes are illustrated in Figure 1. All participants consumed rescue food packages during their SCR. That is, all of them consumed about 250 kcal every 24 h during their fasting. The reasons for consuming rescue food packages included feeling hungry and not wanting to take medication on an empty stomach. Only mild fasting-related symptoms, including feelings of hunger and dizziness, were reported during the first two cycles of SCR. The participants in the SCR group demonstrated a stable nutritional status. Considering only SCR participants with follow-up data, the mean prealbumin level fell within the reference value (17–34 mg/dL) throughout all intervention cycles. The mean phase angle increased gradually from 4.92° ± 0.95 at baseline to 5.33° ± 0.58 at the final cycle of SCR. The mean BMI also increased over time, from 22.50 ± 2.93 at baseline to 23.82 ± 2.68 at the final cycle of SCR (Table 2). Because all participants completed at least two cycles, only data from the baseline and 21 days after SCR cycles 1 and 2 were compared between groups in the following data analysis.

The results of GEE showed that SCR had effects on hematological parameters. Table 3 shows the two models’ results of the influence of SCR on erythrocytes and lymphocytes. Erythrocyte counts increased from 4.10 × 106 ± 0.56 × 106 µL and 4.11 × 106 ± 0.76 × 106 µL at baseline to 4.51 × 106 ± 0.15 × 106 µL and 4.12 × 106 ± 0.72 × 106 µL post-first SCR, and 4.45 × 106 ± 0.29 × 106 µL and 4.15 × 106 ± 0.57 × 106 µL post-second SCR for the SCR group and comparison group, respectively (Table 2). Model one, where we aimed to evaluate the SCR effects on erythrocytes, showed that the SCR group’s increase in erythrocyte counts was significantly greater than that of the comparison group after both the first and second cycles of SCR (*p* < 0.001). After controlling for group, time, stage, and age, the SCR group’s erythrocyte counts were 0.39 × 106 µL and 0.32 × 106 µL higher than the counts of the comparison group after the first and second SCR, respectively.

Leukocyte counts had ups and downs at different time points. For the SCR group, the leukocyte counts increased from baseline (6.60 × 103 ± 1.98 × 103 µL) to post-first SCR (7.05 × 103 ± 2.36 × 103 µL) and then dropped to 5.84 × 103 ± 1.52 × 103 µL post-second SCR. For the comparison groups, the leukocyte counts dropped first from 11.43 × 103 ± 8.21 × 103 µL at baseline to 6.23 × 103 ± 2.18 × 103 µL post-first SCR and increased slightly to 7.48 × 103 ± 2.93 × 103 µL post-second SCR (Table 2). Model two, where we aimed to evaluate the effects of SCR on leucocytes, suggested a significant relationship between SCR and leucocyte counts post-first SCR (*p* < 0.001). The leukocyte count of the SCR group was 5.66 × 103 µL more than that of the comparison group post-first SCR, after controlling for group, time, stage, and age. However, no statistically significant effect was found on leucocytes post-second SCR.

## 4. Discussion

To our knowledge, this is the first study to examine the safety, feasibility, and potential effects of SCR in patients with advanced DLBCL. Our results showed that SCR is safe and feasible, which is comparable to the findings of previous studies [10]. However, most previous studies have only addressed females with gynecological cancer. Our results are particularly important for demonstrating that SCR is also possible in male patients with hematological malignancies. Our preliminary results further suggested that SCR was effective in protecting normal cells or enhancing their regenerative ability due to drug toxicity.

The safety and feasibility of SCR were determined based on 100% retention and compliance rates, minimal fasting-related symptoms, and stable nutrition indicators. The high retention and compliance rates over multiple SCR cycles were more encouraging as compared to those in previous studies. A couple of studies reported over 80% compliance in performing one SCR cycle [11,12]; however, the adherence rate was lower than 50% when followed up for more than three cycles [19,21]. Although the sample size of this study was small, the optimal adherence rate may be secured by our retention strategies, including using a social networking application to establish a channel that allowed researchers and participants to communicate freely. Additionally, as we mentioned in the background, the relatively short period of chemo drug administration may make the adherence to SCR easier than performing SCR with a longer administration of chemotherapeutic drugs. Future studies should examine whether these strategies are effective at enhancing compliance rates in larger samples.

The fasting-related symptoms of hunger and dizziness were mild and only reported during the first two cycles of SCR. While there were several fasting-related symptoms reported in other studies, such as hunger, fatigue, dizziness, and headache, researchers all agree that these symptoms are insignificant [10]. According to participants’ reports of fasting-related side effects in this study, they seemed to adapt to the low calorie intake after the second cycle of SCR. Interestingly, while all participants consumed a rescue food package sometime during their SCR, some patients attributed their action to ‘avoiding taking medication on an empty stomach.’ This reason for consuming food has not been reported in other literature and may be associated with the type of drug and cultural issues. Patients who are on R-CHOP need to take steroids for a period of time and will be told to take steroids with food. From the traditional view of Chinese medicine, it is said that “in the case of illnesses located above the diaphragm, the medications should be taken after meals [40].” It is important to clarify and consider how these factors affect eating and fasting when implementing SCR.

This study is one of the few studies that evaluated the nutritional status during SCR by examining body composition, showing promising results. The prealbumin level, which can reflect short-term changes of nutritional status, remained stable. The phase angle and BMI even increased from the baseline to the last cycle of intervention for patients performing SCR. These results were similar to findings from a study that showed stable body weight over three cycles of 60 h SCR [11]. Compared to BMI, phase angle is a relatively new concept and has been used in many recent studies to estimate nutritional and disease outcomes. Evidence shows that phase angle is a strong predictor of nutritional status, function, quality of life, and survival in various cancer and cardiovascular patients [41,42,43]. While a phase angle greater than 5° is recommended as a cutoff for better outcomes [41,43], the phase angle of patients in the SCR group increased from 4.92° to 5.4°, which showed a clinically meaningful improvement. This improvement is more optimal than a study that observed a slight decrease in phase angle over the four cycles of 96 h SCR [19]. Thus, our results, in line with other studies [10], suggest that a 48 h SCR did not interfere with patients’ nutritional status. Future studies are needed to compare the body composition data between SCR groups and patients receiving regular care.

While more data are needed, the potential protective or regenerative effects of SCR on normal hematological cells were observed in this study. Although we were only able to analyze data before and after two SCR cycles between groups due to the small sample size, our results indicated that participants in the SCR group consistently had significantly higher erythrocyte counts than the comparison groups after controlling for several factors, including disease stage, age, and time. That is, while erythroid regeneration after chemotherapy occurred in both groups of participants, regeneration was more effective or the initial damage from chemotherapy was less severe in the SCR group. Importantly, this increase in erythrocytes seems to be continuous and clinically significant, as, at the post-test, the mean erythrocyte counts of the SCR group fell into the recommended normal range (4.21–5.9 × 106 µL), whereas the mean erythrocyte counts of the comparison groups were lower than the normal range. A similar statistically significant relationship was also observed between the first SCR and leukocyte counts. However, this statistically significant effect of SCR on leukocytes was not observed in the second SCR. While we measured these blood counts approximately 21 days after chemotherapy, a more prominent change in blood counts may be observed if it is measured at 7–14 days post chemotherapy, during possible nadir. The effects of SCR on erythrocytes and leukocytes have also been observed in previous studies. De Groot et al. (2015) examined SCR in breast cancer patients and found a statistically significant increase in erythrocytes at day 7 and 21 after chemotherapy in the SCR group as compared to the control group. Significant changes in leucocytes were not observed. Other studies have reported a non-significant trend of less severe neutropenia [17,18]. Based on this evidence, a potentially positive SCR effect on erythrocytes and leucocytes was suggested. Many factors may cause a failure to find a continuously significant relationship between SCR and leukocytes. For example, it may be limited by the small sample size across studies and the fact that most patients used granulocyte colony-stimulating factor (GCS-F) with dosage variations. Future studies should expand the sample size, control more confounding variables, and examine various indicators, such as side effects, to confirm the protective or regenerative effects of SCR on normal cells.

The inherent limitations of the pilot research are presented in the current study, including the small sample size and retrospective comparison group. Because SCR was never tested in this population, we adopted a more conservative strategy (i.e., physician referral) to recruit patients. In addition, due to the budget limits, we only have about half a year for recruitment. These all contributed to the restricted sample size. Thus, we selected GEE with a model-based estimator, which may have better properties than other analysis methods for small sample sizes [44]. However, future studies with sufficient sample sizes are needed to minimize bias and detect small effects. As no significant differences were found in terms of patient characteristics between groups, the selection strategies for avoiding systematic differences were effective. Randomized controlled trials are the next step to comprehensively examine the effects of the intervention. Another limitation was related to the different baseline. These differences were handled by GEE, which controlled potential confounding variables and focused on the degree of changes.

## 5. Conclusions

This study examined SCR, which requires 48 h of calorie reduction with continuous chemotherapy cycles in an underexplored population and area. The results indicated the safety, feasibility, and positive effects of SCR in patients with DLBCL in Taiwan. Contrary to the recommendations of most in the literature about sufficient calorie intake to prevent cancer cachexia, this study provided a completely different way of thinking—cancer patients may need stage-specific (e.g., treatment phase vs. recovery phase) guidance rather than uniform guidance regarding nutrition. Our results encourage more randomized controlled trials with adequate sample sizes to validate the various short-term and long-term effects of SCR. For clinicians, the SCR might be a potentially effective adjuvant therapy in advanced DLBCL and warrants more attention in order to provide disease-specific and treatment-tailored nutritional interventions.

## Figures and Tables

**Figure 1 nutrients-13-03268-f001:**
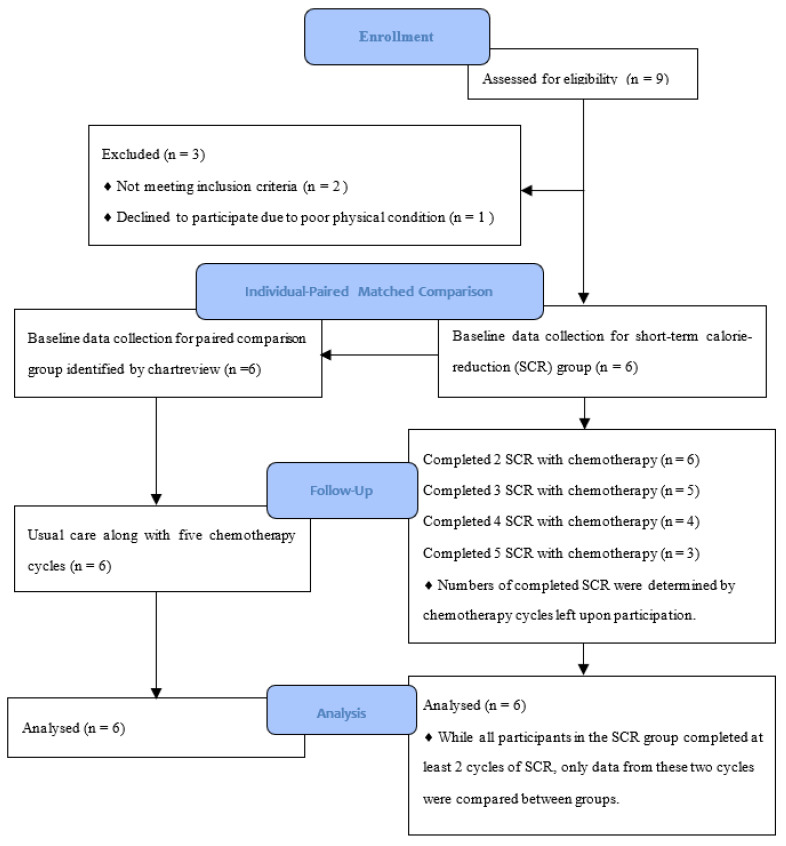
Flow Diagram of Recruitment and Follow-up. SCR, short-term calorie-reduction.

**Table 1 nutrients-13-03268-t001:** Patient characteristics.

	SCR (n = 6)	Comparison (n = 6)	*p* Value
Median Age (range)	60.50 (46–69)	57.50 (50–68)	0.75
Sex: Male	6 (100%)	6 (100%)	
Ethinicity: Chinese	6 (100%)	6 (100%)	
Mean Body Mass Index (SD)	22.50 (2.93)	23.97 (3.87)	0.47
Lugano (modified Ann Arbor) Stage			
III	1 (17%)	3 (50%)	0.39
IV	5 (83%)	3 (50%)	

Note: SCR, short-term calorie reduction.

**Table 2 nutrients-13-03268-t002:** Outcomes of short-term calorie reduction (SCR) and comparison groups before and after intervention.

Outcome Indicators Mean (SD)	SCR Group (*n* = 6)				Comparison Group (*n* = 6)			
	Baseline	First SCR	Second SCR	Final SCR ^a^	Baseline	First SCR	Second SCR	Final SCR
Prealbumin ^b^	27.1 (6.39)	32.54 (3.14)	30.98 (2.14)	27.02 (9.76)	-	-	-	-
Phase angle	4.92° (0.95)	4.64° (0.56)	5.34° (0.76)	5.33° (0.58)	-	-	-	-
Body Mass Index	22.50 (2.93)	23.31 (1.87)	23.69 (2.68)	23.82 (2.68)	23.97 (3.87)	-	-	-
Erythrocytes ^c^	4.10 (0.56)	4.51 (0.15)	4.45 (0.29)	4.3 (0.27)	4.11 (0.76)	4.13 (0.72)	4.15 (0.57)	4.22 (0.52)
Leucocytes ^c^	6.60 (1.98)	7.05 (2.36)	5.84 (1.52)	6.53 (1.53)	11.43 (8.21)	6.23 (2.18)	7.48 (2.93)	5.76 (1.28)

Note: SCR, short-term calorie reduction. ^a^ The final SCR cycle means the second SCR for one participant; the third SCR for one participant; and fourth SCR for one participants; and fifth SCR for three participants. ^b^ The unit for prealbumin is mg/dL. ^c^ The unit for erythrocyte counts is 10^6^/µL and the unit for leucocyte count is 10^3^/µL.

**Table 3 nutrients-13-03268-t003:** Generalized estimating equation (GEE) analysis of erythrocytes and leucocytes.

Variables	Model 1: Erythrocyte Counts ^a^			Model 2: Leucocyte Counts ^a^		
	Estimate	SE	*p* Value	Estimate	SE	*p* Value
Intercept	6.93	3.64	0.06	21.07	10.89	0.05
SCR group ^b^	0.13	0.41	0.74	−4.40	2.56	0.085
Cycle 1 ^c^	0.02	0.003	0.000 **	−5.20	0.64	0.000 **
Cycle 2 ^d^	0.04	0.003	0.000	−3.94	2.08	0.058
Stage	−0.24	0.54	0.65	−0.64	1.59	0.69
Age	−0.03	0.04	0.34	−0.13	0.11	0.23
SCR group × SCR cycle 1 ^e^	0.39	0.005	0.000 **	5.66	0.91	0.000 **
SCR group × SCR cycle 2 ^f^	0.32	0.005	0.000 **	3.19	2.94	0.28

Note: SCR, short-term calorie reduction. ^a^ The unit for erythrocyte counts was 10^6^/µL and the unit for leucocyte count was 10^3^/µL, ^b^ Reference group: comparison group, ^c^ Reference group: baseline, ^d^ Reference group: cycle 1, ^e^ Reference group: group (comparison) × SCR cycle (baseline), ^f^ Reference group: group (comparison) × SCR cycle (cycle 1), ** indicated *p* < 0.001.

## Data Availability

The data presented in this study are available on request from the corresponding author. The data are not publicly available due to patient privacy.

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
