# Peer review of "Safety, Feasibility, and Effects of Short-Term Calorie Reduction during Induction Chemotherapy in Patients with Diffuse Large B-Cell Lymphoma: A Pilot Study"

_nutrients, 2021, doi:10.3390/nu13093268_

Round 1

Reviewer 1 Report

This nice paper shows the beneficial effect of short-term calorie reduction (possibly causing less tumor lysis syndrome?), which is an interesting but little-used nutritional technique. I have some remarks, though:

  • Despite the results, most studies claim the opposite, i.e. that an intensive nutritional intervention is needed, rather than caloric restriction, and the authors should comment upon this.
  • Water deprivation is dangerous in patients receiving chemotherapy (especially in Cisplatinum-based regimens), and the authors should address this issue.
  • Was the evolution of the thrombocyte number assessed?

Reviewer 2 Report

This article represents a first study investigating a short-term calorie restriction/fasting regime in patients with DLBCL and one of the first in hematological cancer patients, and aims to address concerns about the translation of such a diet in terms of gender, ethnicity and cancer types. Although the content and results of the study are promising, the manuscript needs major revision to be reconsidered for publication in Nutrients. 

Major revisions: 

  1. The materials and methods are too brief and lack details: please include details about: patient characteristics; how the cell phase angle is performed; the R-CHOP details (milligrams of dexamethason, adherence)
  2. Only 6 patients were included; although it is a pilot study, is this number of patients based on any previous finding? why did you choose 6? 
  3. how many patients with DLBCL in advanced stage are seen at your outpatient clinic a year? is it true only 9 patients were seen in the period of inclusion? 
  4. Did you consider biochemical parameters (like albumin, retinol binding protein) to evaluate nutritional status before and after diet? 
  5. Please provide information of the nutritional contents of the food packages the patients received and took; also, how much % of caloric restriction was received including these packages?
  6. These six patients received 2 to 5 cycles of caloric restriction; you only evaluated the first 2 cycles of fasting. Which cycle were all patients in when you started the dietary regime? do you not think previous chemotherapies affect the outcome, like the baseline erytrocytes and leukocytes? 
  7. Patient characteristics table: include male/female ratio, ethnicity etc. If you do not have these data or do not want to provide it, please remove this research question from you introduction and discussion since you do not have any results that provide evidence to answer this question in the current state of the manuscript. 
  8. The changes in erytrocytes and leukocytes are small and do not appear clinically relevant. Please try to relate the outcome to a clinical setting; also, changes in these parameters do not provide evidence for reducing drug toxicity per se, especially since any other evidence for drug toxicity reduction is missing, such as reports of side effects, neutrophile counts, reticulocytes, complications, etc. Please nuance these statements in your discussion or remove it.
  9. Also, leukocyte levels of the SCR and comparison group are different and the baseline levels of the comparison group is even too high; did these patients have infections? other reasons why the levels were higher? was their complete compliance to the R-CHOP in both groups?
  10. Can you explain why BMI levels increased during the chemotherapy cycles, even though the patients received severe caloric restriction during several days? were they overcompensation of food intake in other days? do you have insight into the food diaries that could explain these results?

Minor revisions: 

- Grammar check; In the introduction, please correct R-COHP into R-CHOP (line 96). 

Round 2

Reviewer 2 Report

The authors have extensively edited the paper after the major revisions and the manuscript would now be ready for publications.